Using InVEST to evaluate water yield services in Shangri-La, Northwestern Yunnan, China

Yu Yuanhe 1 2
Sun Xingqi 3
Wang Jinliang 1 jlwang@ynnu.edu.cn
Zhang Jianpeng 1
1 Faculty of Geography, Yunnan Normal University , Kunming, Yunnan , China
2 School of Geography, Nanjing Normal University , Nanjing, Jiangsu , China
3 The First Geodetic Surveying Brigade of MNR , Xi’an, Shaanxi , China
Provete Diogo
Electronic publication date: 2022 Jan 14
Publication date: 2022
Volume: 10
Electronic Location ID: e12804
Received 2021 Oct 25; Accepted 2021 Dec 27
Copyright: © 2022 Yu et al.
Copyright year: 2022
Copyright holder: Yu et al.
License: This is an open access article distributed under the terms of the Creative Commons Attribution License, which permits unrestricted use, distribution, reproduction and adaptation in any medium and for any purpose provided that it is properly attributed. For attribution, the original author(s), title, publication source (PeerJ) and either DOI or URL of the article must be cited.
License URL: https://creativecommons.org/licenses/by/4.0/

Keywords: Water yield, InVEST model, Ecosystem services, Shangri-La City, Northwest Yunnan, China

Funding: “Natural Forests Biomass Estimation at Tree Level in Northwest Yunnan by Combining ULS and TLS Cloud Points Data” 41961060 “Environmental monitoring and assessment of land use/land cover change impact on ecological security using geospatial technologies” 2018YFE0184300 Program for Innovative Research Team (in Science and Technology) in the Universities of Yunnan Province, IRTSTYN This research was supported by the National Natural Science Foundation of China for the “Natural Forests Biomass Estimation at Tree Level in Northwest Yunnan by Combining ULS and TLS Cloud Points Data”, grant number 41961060; by the Multi-government International Science and Technology Innovation Cooperation Key Project of National Key Research and Development Program of China for the “Environmental monitoring and assessment of land use/land cover change impact on ecological security using geospatial technologies”, grant number 2018YFE0184300; by the Program for Innovative Research Team (in Science and Technology) in the Universities of Yunnan Province, IRTSTYN. The funders had no role in study design, data collection and analysis, decision to publish, or preparation of the manuscript.

==============================
Water yield is an ecosystem service that is vital to not only human life, but also sustainable development of the social economy and ecosystem. This study used annual average precipitation, potential evapotranspiration, plant available water content, soil depth, biophysical parameters, Zhang parameter, and land use/land cover (LULC) as input data for the Integrated Valuation of Ecosystem Service Tradeoffs (InVEST) model to estimate the water yield of Shangri-La City from 1974 to 2015. The spatiotemporal variations and associated factors (precipitation, evapotranspiration, LULC, and topographic factors) in water yield ecosystem services were then analyzed. The result showed that: (1) The water yield of Shangri-La City decreases from north and south to the center and showed a temporal trend from 1974 to 2015 of an initial decrease followed by an increase. Areas of higher average water yield were mainly in Hutiaoxia Town, Jinjiang Town, and Shangjiang Township. (2) Areas of importance for water yield in the study area which need to be assigned priority protection were mainly concentrated in the west of Jiantang Town, in central Xiaozhongdian Town, in central Gezan Township, in northwestern Dongwang Township, and in Hutiaoxia Town. (3) Water yield was affected by precipitation, evapotranspiration, vegetation type, and topographic factors. Water yield was positively and negatively correlated with precipitation and potential evapotranspiration, respectively. The average water yield of shrubs exceeded that of meadows and forests. Terrain factors indirectly affected the ecosystem service functions of water yield by affecting precipitation and vegetation types. The model used in this study can provide references for relevant research in similar climatic conditions.

Introduction

Ecosystem services are direct or indirect benefits provided by nature to mankind (IPBES, 2019; Zhan et al., 2019). Ecosystem services generally include exported tangible material products and provide intangible services, which can be further categorized into providing, supporting, regulating, and cultural and entertainment services (Rijal et al., 2021; Sun, 2017). The rapid growth of the global population, economic development, improvement of living standards, and increasing human water demand have resulted in the elevation of water shortages as a major threat to the sustainable development of human society (Huang et al., 2021).

As an ecosystem service, water yield is not only an important foundation for maintaining regional biodiversity and key ecosystem functions, but also affects the development and distribution of the regional population, society, and economy (Sharafatmandrad & Khosravi Mashizi, 2021; Zhao et al., 2019). Therefore, water shortages may lead to a variety of social and environmental problems, including drinking water deficits, reductions to crop yield, and water quality degradation (Huang et al., 2021). Water yield is an important parameter describing regional water resources (Li et al., 2021a) and also plays an indispensable role in other ecological functions, such as the carbon cycle and sediment transport (Ajaz Ahmed et al., 2017; Wang et al., 2020). Therefore, the study of water yield is helpful for understanding the evolution of the water yield ecosystem service function and for exploring the relationship between human beings and water resources, which has great significance for the scientific management and rational utilization of water resources (Dennedy-Frank et al., 2016; Lian et al., 2020).

There remains no established academic definition of water yield as an ecosystem service. The use of the term ‘water yield’ mainly relates to the ability of precipitation to directly and effectively replenish surface water and groundwater within a certain period of time, thereby enhancing human well-being (Fan, Shibata & Chen, 2018; Li et al., 2021a). Calculation of the water yield remains a complex process since the water yield is affected not only by rainfall intensity, soil permeability, slope, and vegetation, but also by climate and land use/land cover (LULC) (Wei et al., 2021; Yin et al., 2020). Climate change affects water yield by influencing precipitation and evaporation in the watershed. The LULC can affect the water yield by changing the water cycle of the watershed, thereby affecting evapotranspiration, the infiltration process, and water retention (Song et al., 2015). Past attempts to calculate the water yield have utilized measured data such as precipitation storage and soil water storage capacity within water balance methods (Li, Xie & Xiao, 2009).

The recent rapid development of remote sensing technology and hydrological models has prompted many scholars to quantitatively, visually, and finely analyze and evaluate the water yield function of a regional ecosystem through model simulation. Hydrological models, such as the MIKE System Hydrological European (MIKE SHE) (Refsgaard, Storm & Singh, 1995), Soil and Water Assessment Tool (SWAT) (Cong et al., 2020), Water Supply and Stress Index (WaSSI) (Ajaz Ahmed et al., 2017), Variable Infiltration Capacity (VIC) (Srivastava, Kumari & Maza, 2020), and Integrated Valuation of Ecosystem Services and Tradeoffs (InVEST) (Yang et al., 2019a) have shown good performance under various geographic conditions. The InVEST mode was developed in 2007 through a collaboration between Stanford University, the World Wildlife Fund, the Nature Conservation Society, and other related institutions on the basic of the Budyko curve and annual precipitation (Yang et al., 2019a). The model is relatively simple, suited for applications to catchments lacking observed datasets (Lian et al., 2020; Wei et al., 2021) and provides a reliable method for estimating water yield under different spatiotemporal scales (Yang et al., 2019a; Yin et al., 2020). Furthermore, the InVEST model calculates water yield based on the water balance to fully consider the spatial differences in soil permeability under different LULC and the influencing factors, such as topography and runoff. The model allows the quantitative estimation of the water yield in different landscape types and the results can be reported in the form of shapefiles, tables, and grid maps (Li et al., 2021b).

The advantages of the InVEST water yield model include strong spatial expression, dynamic evaluation capabilities, simple input data requirements, large output data volume, and strong scalability. Therefore, the model is widely used to: (1) evaluate the impact of land use on water-related ecosystem services (Gao et al., 2017); (2) explore the impact of urbanization on water yield (Xu et al., 2019); (3) analyze the effects of climate, topography, soil, land use types, and other factors on water yield (Dai & Wang, 2020; Li et al., 2021b; Nahib et al., 2021). The above studies demonstrated that the InVEST water yield model can reliably estimate the water yield ecosystem service in a wide range of catchment types. However, the water yield ecosystem service function varies widely among different regions due to differences in responses to precipitation, evapotranspiration, LULC, and topography (Xie et al., 2017). Few previous studies have attempted to examine the effects of these climatic factors, land use patterns, and topographic factors on water yield in alpine-gorge areas with a complex topography, diverse climate types, and a fragile ecological environment (Luo et al., 2021).

Shangri-La City, northwestern Yunnan, China, is a typical alpine-gorge area with a complex topography, large extents of sloped cultivated land, and reclamation of steep sloped areas. Therefore, the area is prone to soil erosion, geological disasters, and ecological problems (Sun, 2017). Improper exploitation of forest resources has resulted in the extinction of a large number of species, decline of soil fertility, and loss of biodiversity in some areas of Shangri-La (Yang et al., 2019b). The land-use changes in the region would be bound to affect the water yield by influencing the hydrological cycle. Climate change increases the uncertainty and complexity of the hydrological cycle, thereby changing the water yield service to a certain extent. In addition, Shangri-La City is facing numerous challenges that urgently need to be resolved, including serious water resource shortages, an uneven spatiotemporal distribution of water resources, an unsustainable water supply structure, and extensive exploitation of water resources (Li & Wang, 2016). Therefore, as a typical alpine-gorge area, there is an urgent need to explore the temporal and spatial differentiation in water yield of in Shangri-La City, northwestern Yunnan.

The aims of this study included: (1) Estimation of the water yield of Shangri-La City, northwestern Yunnan Province from 1974 to 2015; (2) classification of the importance of water yield ecosystem service functions; (3) exploration of the responses of water yield to climate factors, LULC, and topographic factors. The results of this study are expected to provide a reference for the study of water yield, water resources management, and ecological security in Shangri-La City and similar ecosystems.

Materials and Methods

Study area

Shangri-La City is in the northwestern part of Yunnan Province, China, between 26°52′–28°52′N and 99°20′–100°19′E and has a total area of 11,600 km2 (Fig. 1). The city forms a core part of the World Natural Heritage site “Three Parallel Rivers” and falls within a fragile and sensitive area of the ecosystem in the hinterland of the Hengduan Mountains. The city has also been designated a biodiversity hotspot of global significance by the World Wildlife Fund (WWF) (Yang & Wang, 2015). The climate of Shangri-La City is affected by the monsoon originating from the southwestern Indian Ocean. The wet and dry seasons extend from June to October and November to May, respectively. Shangri-La City has a complex and diverse topography with the local climate characterized by high spatial variability (Liu et al., 2021b). Other climate characteristics of the region include a high atmospheric transparency, increased daytime solar radiation, and rapid increases in temperature, resulting in a large diurnal temperature range. The Jinsha River is the dominant river system in Shangri-La City, with the 13 main tributaries being the Shuoduogang, Dongwang, Gangqu, Jiren, Langdu, Liangmei, Annan, Tangman, Anle, Maidi, Niru, Xinglong, and Baishui rivers. The average surface water and groundwater resources in the study area over many years are 5 billion m3 and 2 billion m3, respectively (Li & Wang, 2016). Shangri-La City is rich in soil resources, mainly dark brown soil, brown soil, red soil, alpine shrub meadow soil, brown dark coniferous forest soil, alpine cold desert soil, and alluvial soil (Chen et al., 2019). Most areas of Shangri-La City are subalpine and alpine landforms with elevations exceeding 3,000 m and the region contains 43 species of major forest trees, including 10 species of coniferous forests and 33 species of broadleaf forests (Liu et al., 2018; Yu et al., 2019; Zhang, Wang & Liu, 2020).

Figure 1 The geographical location of Shangri-La City.

(A) Map of Yunnan Province showing the position of Shangri-La City; (B) a digital elevation model (DEM) of Shangri-La City.

The InVEST water yield model

Water yield refers to the amount of water produced per unit area over a given period. The InVEST water yield model runs on a gridded map and is an estimation method based on the Budyko hydrothermal coupling equilibrium hypothesis in which water yield in an area is obtained by subtracting the total actual evapotranspiration per unit area from total precipitation (Budyko, 1974). The equations describing the water yield are:

(1) Y(x)=(1−AET(x)P(x))×P(x)

where Y(x), AET(x), and P(x) are the total annual water yield (mm), the annual actual evapotranspiration (mm), and the average annual precipitation (mm) of grid unit x, respectively. AET(x)P(x) is the ratio of actual evapotranspiration to average annual precipitation (Zhang et al., 2004) and is calculated as:

(2) AET(x)P(x)=1+PET(x)P(x)−[1+(PET(x)P(x))w]1/w

(3) PET(x)=Kc(x)×ET0(x)

(4) w(x)=AWC(x)×ZP(x)+1.25

(5) AWC(x)=min (MaxSoilDepthx,RootDepthx)PAWCx

In Eqs. (2) to (5), PET(x) is the potential evapotranspiration of grid unit x, Kc(x) is the reference crop evapotranspiration coefficient, ET0(x) is the reference crop evapotranspiration (mm d−1), w(x) is a nonphysical empirical fitting parameter of natural climate-soil properties, AWC(x) is the available water content of plants and is used to determine the amount of water stored and released by the soil for vegetation growth, calculated as the difference between the field water holding capacity and the wilting point, Z is the Zhang parameter, MaxSoilDepthx is the maximum soil depth, and RootDepthx is the root depth.

Correlation coefficient (R) and significance test

The Pearson correlation coefficient (R) is a measurement of the degree of linear correlation between variable, and used to quantify the correlation between water yield and precipitation and between water yield and PET for Shangri-La City from 1974 to 2015 at a pixel level. An R ≥ 0 indicates a positive correlation, whereas an R < 0 indicates a negative correlation, and the closer the absolute value of R is to 1, the stronger the correlation.

Data sources and processing

This study integrated remote sensing, a geographic information system (GIS), and the InVEST model. The data used in the study included precipitation, potential evapotranspiration, plant available water content, soil depth, land use/land cover (LULC) data, the biophysical parameters, and the Zhang parameter. In addition, the ENVI 5.3 (https://www.l3harrisgeospatial.com/Software-Technology/ENVI), ArcGIS 10.5 (https://www.esri.com/en-us/arcgis/products), SPSS 22 (https://www.ibm.com/products/spss-statistics), and InVEST (https://naturalcapitalproject.stanford.edu/software/invest) software packages were used in the study to reprocess and analyze data. The spatial resolution of all input data used in the InVEST model were unified to 30 m × 30 m.

Precipitation (P) and reference evapotranspiration ( ET0 )

Meteorological data, including precipitation, temperature, and solar radiation, were provided by the Institute of Geographic Sciences and Natural Resources of the Chinese Academy of Sciences (http://www.igsnrr.ac.cn/xxgx/200909/t20090904_2464203.html), Yunnan Meteorological Services (http://yn.cma.gov.cn/), and the Meteorological Administration of China (https://data.cma.cn/, last accessed March 20, 2021). The meteorological data were obtained from 13 meteorological stations distributed across the study area and as well as from the surrounding meteorological stations, Shangri-La, Lijiang, Weixi, Ninglang, Deqin, Gongshan, Fugong, Lanping, Jianchuan, Heqing, Yongsheng, Huaping, and Eryuan. The daily meteorological data were subjected to sorting, interpolation, and clipping preprocessing to obtain annual-scale meteorological data in the standard format of the InVEST model. The kriging interpolation method in ArcGIS 10.5 was used to create a precipitation map of the study area.

The map of annual reference evapotranspiration was generated based on the extraterrestrial radiation, temperature, and monthly average precipitation. The modified Hargreaves method was used to estimate the ET0 in areas with a lack of observed data (Droogers & Allen, 2002), with estimates shown to be better than those of the Penman-Monteith method (Nahib et al., 2021). The monthly and annual reference evapotranspiration maps were obtained through the ArcMap raster calculator, calculated as follows:

(6) ET0=0.0013×0.408×Ra×(Tavg+17)×(TD−0.0123P)0.76

In Eq. (6), Ra is the extraterrestrial radiation (MJ m−2 d−1), represented in the this study as half of the total solar radiation of the meteorological station, Tavg is the mean value of the average highest and average lowest temperatures (°C), TD represents the difference between the average highest temperature and the average lowest temperature (°C), and P is the average monthly precipitation (mm).

Plant available water content (PAWC) and soil data

The plant available water content (PAWC) parameter is used to evaluate the total water stored and released by soil for plants and is determined by the mechanical composition of the soil and the root depth of the vegetation, calculated as per Zhou et al. (2005) as:

(7) PAWC=54.509−0.132sand−0.003(sand)2−0.055silt−0.006(silt)2                   −0.738clay+0.007(clay)2−2.688OM+0.501(OM)2

In Eq. (7), sand, silt, clay, and OM are the contents of sand (%), silt (%), clay (%), and organic matter (%) in the soil, respectively. According to the model input requirements, the PAWC values were converted to a fraction from 0 to 1.

The soil data were collected from the field and the data of the second soil survey were provided by the soil and fertilizer station of the Diqing Tibetan Autonomous Prefecture, including soil species, soil-forming parent materials, soil structure, soil depth, soil profile, detailed soil mechanical composition, and soil organic matter content. Field data were collected based on the soil genetic layer, selecting the relatively original sites less affected by human activities, and soil samples of each occurrence layer were collected from bottom to top. The soil organic matter content was estimated by multiplying 1.724 with the soil organic carbon content that was measured by potassium dichromate oxidation with external heating (MOA, 2006).

Land use/land cover (LULC)

This study used LULC data for 1974, 1989, 2003, and 2015, with the data for the first three years were the acceptance data of the Nature Conservancy (http://www.tnc.org.cn/) project entitled “30 Years Land Use/Land Cover Change (LUCC) of Shangri-La County (2004-2006)” undertaken by Key Laboratory of Resources and Environmental Remote Sensing for Universities in Yunnan (http://www.ynnurs.com/). The LULC data for 2015 were obtained by visual interpretation and modified based on the classification data in 2003 and the Landsat OLI data in 2015 (downloaded from https://glovis.usgs.gov/, last accessed March 20, 2021). The accuracy of the LULC data was verified through field survey data and was shown to be 94%. The LULC data were classified into 27 classes with a spatial resolution of 30 m.

Biophysical parameters

Table 1 shows the biophysical parameters of each LULC class used in the model. The data for root depth were set by reference data from InVEST model documents and with reference to the report: “Crop evapotranspiration: Guidelines for computing crop water requirements”, published by the Food and Agriculture Organization of the United Nations (FAO) (https://www.fao.org/3/X0490E/x0490e00.htm#Contents). Different root depths were set for the different land use types. Kc was set according to the reference value of the evapotranspiration coefficient of the FAO.

Table 1 Biophysical parameters used in the InVEST water yield model applied to Shangri-La City, Yunnan Province, China.

Lucode	LULC_desc	Kc	Root_depth (mm)	LULC_veg	
1	Urban land	0.10	10	0	
2	Rural residential land	0.30	500	0	
3	Highway land	0.20	100	0	
4	Airport	0.10	10	0	
5	Terrace	0.65	2,000	1	
6	Flat field land	0.65	2,000	1	
7	Flat dryland	0.60	1,000	1	
8	Slope dryland	0.60	1,000	1	
9	Orchard	0.70	3,000	1	
10	Deciduous broad-leaved forest	1.00	7,000	1	
11	Evergreen broad-leaf forest	1.00	7,000	1	
12	Coniferous forest	1.00	7,000	1	
13	Mixed forest	1.00	7,000	1	
14	Alpine shrub	0.50	2,000	1	
15	Subalpine shrub	0.50	2,000	1	
16	Valley shrub	0.60	2,000	1	
17	Alpine meadow	0.65	1,700	1	
18	Subalpine meadow	0.65	1,700	1	
19	Valley grassland	0.40	650	1	
20	Inland wetland	0.60	1,000	1	
21	River	0.00	1,000	1	
22	Lake	0.00	1,000	1	
23	Reservoir	0.00	1,000	0	
24	Bare land	0.00	200	1	
25	Bare rock	0.00	200	1	
26	Flood land	0.00	200	1	
27	Permanent snow glacier	0.00	500	1	
Notes:

Lucode: LULC code for each type.

LULC_desc: description of each type of LULC name.

Kc: plant evapotranspiration coefficient for each LULC type.

Root_depth: the maximum root depth for each LULC type.

LULC_veg: LULC attribute category.

The Zhang parameter

The Zhang parameter is an important input parameter used to estimate water yield in the InVEST model and characterizes the seasonal distribution of precipitation within a range of [0, 10]. The Zhang parameter was set to 3.8 after several test corrections as being closest to the results published in the Shangri-La Water Resources Bulletin.

Results

Analysis of the InVEST model input data

The input data of the InVEST water yield model included annual precipitation, potential evapotranspiration (PET), plant available water capacity (PAWC), soil depth, and LULC. The average precipitation values in each region of Shangri-La City in 1974, 1989, 2003, and 2015 were 777.52 mm, 865.32 mm, 823.84 mm, and 733.48 mm, respectively, whereas the average PET values were 1,100.80 mm, 973.01 mm, 1,026.06 mm, and 1,024.88 mm, respectively. The input data for 2015 were taken as a case study to analyze the changes in the above parameters (Fig. 2). As shown in Fig. 2A, precipitation in 2015 varied spatially from 598.33 mm to 881.51 mm. The precipitation showed a highly uneven spatial distribution, with high rainfall in the south, low rainfall in the central regions, and slightly higher rainfall in the north. The annual rainfall values in Jinjiang and Hutiaoxia in 2015 were relatively high at 859.63 mm and 832.87 mm, respectively. Fig. 2B shows that the annual PET in 2015 ranged from 923.63 mm to 1,194.55 mm across the different regions. The annual PET values of Jinjiang and Hutiaoxia were also relatively high at 1,137.14 mm and 1,130.47 mm, respectively. The PET decreased gradually from south to north, showing a generally positive relationship to precipitation. Fig. 2C shows that the PAWC decreased with decreasing elevation, with high PAWC mainly distributed in areas such as Gezan, Dongwang, and Wujing. Conversely, soil depth decreased with increasing elevation, with low soil depth mainly distributed in areas such as Geza, Jiangtang, and Dongwang (Fig. 2D). The LULC in 2015 mainly comprised mixed forest, coniferous forest, alpine shrub, and alpine meadow, accounting for 35.86%, 32.17%, 8.96%, and 8.45% of the total area of the study area, respectively (Fig. 2E).

Figure 2 Spatial distributions of the InVEST model input data for 2015 for application to Shangri-La City, Yunnan Province, China.

(A) Annual precipitation (P); (B) annual potential evapotranspiration (PET); (C) plant available water content (PAWC); (D) soil depths; (E) land use/land cover (LUCC). Abbreviations: LJ, Luoji; DW, Dongwang; NX, Naxi; GZ, Geza; WJ, Wujing; SJ, Shangjiang; JJ, Jinjiang; JT, Jiantang; XZD, Xiaozhongdian; SB, Sanba.

Characteristics of spatiotemporal variation of water yield

Figure 3 shows that water yield was generally higher in the north and south and lower in the central region. The ranking of the different simulation years in terms of spatially averaged annual water yield for all villages and towns was 1989 > 2003 > 1974 > 2015, with water yields of 316.05, 266.42, 221.27, and 199.06 mm, respectively. Except Dongwang Township which showed the largest annual water yield in 2003 of 314.68 mm, the annual water yields of other townships were in general consistent with that of Shangri-La City. The average water yields in Hutiaoxia, Jinjiang, and Shangjiang across the different years were relatively high. Except in 1989, the annual water yields of Jiantang Town, Wujing Township, and Luoji Township were relatively low. Taking Gezhan and Dongwang as examples in Fig. 2, the highest water yields were in areas with high PAWC, low soil depth, and alpine meadows.

Figure 3 A summary of water yield for Shangri-La City, Yunnan Province, China as simulated by the Integrated Valuation of Ecosystem Service Tradeoffs (InVEST) model.

The spatial distribution of water yield for different years is shown in (A), (B), (C), and (D), whereas (E) shows the spatial distribution of the change in water yield between 1974 and 2015. A statistical summary of average water yield for different areas and different years is shown in (F).

The changes in water yield were classified as a significant decrease (>−30 mm), a decrease (−30 to −10 mm), a non-significant change (−10 to 10 mm), an increase (10–30 mm), and a significant increase (>30 mm). According to the above categorization, the average water yield from 1974 to 2015 from the southeast to the northwest showed significant decreases, decreases, non-significant changes, increases, and significant increases across areas of 3,840, 2,619, 2,584, 924, and 1,646 km2, accounting for 33.0%, 22.6%, 22.3%, 8.0%, and 14.1% of the total area of Shangri-La City, respectively (Fig. 3E).

In addition, the correlations between precipitation and water yield were mainly positive, with an average R of 0.83 and with the area with R ≥ 0.8 accounting for ~77.56% of the total study area (Fig. 4A). In contrast, the correlations between PET and water yield were mainly negative (Fig. 4B), with an average R of −0.52 and the area with R ≤ −0.6 accounting for ~63.61% of the study area. Figure 4C shows the significance of correlation between water yield and precipitation, with the results indicating that 71.48% of the study area passed the significance test of P≤0.1 (including P = 0.01, P = 0.05 and P = 0.1). Areas showing a highly significant correlation (P = 0.01) accounted for 15.36% of the total area, with these areas mainly distributed in the central areas of Shangri-La City, whereas the areas showing significant correlations of P = 0.05 accounted for 44.82% of the total area and were widely distributed. Figure 4D shows the spatial distribution of the significance of the correlations between water yield and potential evapotranspiration, indicating that only 4.534% of the study area passed the significance test of P≤0.1, with these areas sporadically distributed in each region.

Figure 4 Correlation coefficient (R) and significance test between water yield and precipitation (P), and between water yield and potential evapotranspiration (PET) from 1974 to 2015 in Shangri-La City, Yunnan Province, China.

(A) The R between water yield and P; (B) the R between water yield and PET; (C) the significance test of the R between water yield and P; (D) the significance test of the R between water yield and PET.

Spatial division of water yield importance

This study determined the grades of importance of the water yield ecosystem service functions in Shangri-La City according to the “China Ecological Protection Red Line–Ecological Function Red Line Demarcation Technical Guide (trial)”: (Grade I) generally important area (<100 mm); (Grade II) moderately important area (100–200 mm); (Grade III) highly important area (200–300 mm) and; (Grade IV) extremely important area (≥300 mm). Taking 2015 as an example (Fig. 5), the areas of Shangri-La City falling within these different grades in decreasing order were: Grade II (5,053 km2), Grade III (2,427 km2), Grade I (2,349 km2), and Grade IV (1,802 km2), accounting for 43.36%, 20.90%, 20.22%, and 15.52% of the total area, respectively. Areas falling in Grade I were mainly distributed in the east and west of Jiantang, Gezan, and Xiaozhongdian. Those in Grade II were mainly distributed in Naxi, Luoji, the northern part of Sanba, and the western part of Gezan. Those in Grade III were mainly distributed in central Dongwang, Jinjiang, Hutiaoxia, and parts of Sanba and Naxi. Areas falling in Grade IV were mainly distributed to the west of Jiantang, central Xiaozhongdian, central Gezhan, and northwestern Dongwang and Hutiaoxia. Since the areas falling within Grade IV have the highest water yield function value, these areas should be prioritized for protection.

Figure 5 The spatial distribution of water yield in Shangri-La City, Yunnan Province, China as simulated by the Integrated Valuation of Ecosystem Service Tradeoffs (InVEST) model.

(A) According to Grades I–IV indicating the importance of the water yield ecosystem service, (B) the area proportion of each grade in 2015. The grading classification is shown in “Spatial division of water yield importance”.

Discussion

Effect of precipitation and evapotranspiration on water yield

Changes in climatic factors affect water yield mainly by affecting precipitation and PET (Li et al., 2021b). The water yield of Shangri-La City generally showed significant changes corresponding to changes in precipitation and evapotranspiration, with the effect of precipitation on water yield far exceeding that of PET. Many past studies have similarly shown that a decrease in precipitation results in corresponding decreases in water yield (Lian et al., 2020) and that precipitation is considered to be an important factor affecting changes in water yield (Li et al., 2021b; Yang et al., 2021). Evapotranspiration is a key component of the terrestrial water cycle connecting the atmosphere, vegetation, and soil, and represents the sum of vegetation and soil evaporation (Yin et al., 2021).

Effect of LULC on water yield

LULC change is an important environmental issue that affects ecosystem services, and particularly water ecosystem services (Liu et al., 2021a). LULC can change the hydrological flux due to changes in the physical properties of the catchment surface, soil, and vegetation, including properties such as surface roughness, albedo, infiltration capacity, root depth, vegetation index, and stomatal conductance (Wei et al., 2021). The results indicated that the average water yield of shrubs (alpine, subalpine, and valley shrubs) exceeded that of meadows (alpine meadows, subalpine meadows, and valley grassland) and forests (deciduous broad-leaved, evergreen broad-leaf, coniferous, and mixed forests). Subalpine shrubs obtained the highest average water yield over 1974, 1989, 2003, and 2015 of 356.70, 525.27, 407.27, and 327.98 mm, respectively (Fig. 6A). Forest land obtained the lowest average water production since the growth of natural forest land requires the consumption of relatively large quantities of water, the evapotranspiration coefficient of natural forest land is relatively large, and the evapotranspiration capacity of forest is relatively strong. Although shrubland requires more water than grassland, the evapotranspiration coefficient of shrubland is lower than that of grassland. Therefore, the average water yield capacity of shrubland exceeded that of grassland. The average water yield of all natural/semi-natural vegetation types over time first increased and then decreased, reaching a peak in 1989.

Figure 6 The distribution of water yield at different levels of effect factors in Shangri-La City from 1974 to 2015.

(A) and (B) are average water yield and total water yield under natural/semi-natural vegetation types, respectively, (C) and (D) are average water yield and total water yield across different elevations, respectively, (E) and (F) are average water yield and total water yield at different slope intervals, respectively.

There are differences in evapotranspiration, litter water retention, soil water content, and canopy interception among different vegetation types. Correspondingly, the different vegetation types show obvious differences in water yield capacity (Nahib et al., 2021; Yang et al., 2021). Figure 6B shows the total water production among different vegetation types, indicating that although the average water yield of forests is relatively small, the forest area plays a major role in the water yield ecological service function in Shangri-La City given that forest occupies the largest area. In general, forest land showed the highest total water yield across the four periods of 1974, 1989, 2003, and 2015, accounting for approximately 59.69%, 61.80%, 58.21% and 55.75% of the water yield of all vegetation types, respectively. The results of this study showed that LULC is an important factor regulating changes in the water yield ecosystem service function, similar to the conclusion of Guo et al. (2021).

Effect of topographic factors on water yield

Elevation significantly affects the distribution of precipitation and vegetation types, thereby indirectly affecting the water yield capacity. Figs. 6C and 6D shows the effect of elevation on the water yield ecosystem services in Shangri-La City. The average water yield of each year initially decreased and then increased with increasing elevation. Except in 1989, the lowest average water yield occurred at an elevation range of 3,500–4,000 m, whereas the highest average water yield was distributed at an elevation of <2,000 m and the second highest water yield occurred at an elevation exceeding 4,000 m (Fig. 6C). Over the four periods, the highest water yields were mainly concentrated in areas with elevations exceeding 3,000 m, accounting for >80% of the total water yield. In contrast, relatively small water yields were evident at elevations <2,500 m, accounting for ~8.5% of the total water yield (Fig. 6D). There was an increasing trend followed by a decreasing trend in average water yield from 1974 to 2015. The change in average water yield with elevation was greatest in 1989, followed by 2003. Areas of low elevation are mainly distributed in the southern part of the study area characterized by precipitation far exceeding that in the central parts and areas with higher elevation, which results in these areas of low elevation having relatively high average water yield. The proportion of forest area decreased sharply in areas of extremely high elevation, resulting in weakened evapotranspiration and a corresponding increase in the average water yield.

Water yield initially gradually decreased and then gradually increased with increasing slope, with a minimum water yield in a slope range of 35° to 45° (Fig. 6E). The total water yield was affected by the area of each slope grade, with the total annual water yield from 1974 to 2015 showing a normal distribution across the different slope grades (Fig. 6F). The majority of water yield was concentrated between slope values of 5° to 45°, accounting for ~85% of the study area, whereas the remaining slope intervals contained a negligible water yield (Fig. 6F). In addition, within each slope grade, water yield initially increased and then gradually decreased over time, with a peak in 1989. In general, topographical factors indirectly affect the water yield ecosystem service function by affecting precipitation and vegetation types, as also reported by Lian et al. (2020).

Limitations and uncertainty of the model

The InVEST model simplifies the hydrological process in that it does not distinguish between surface and subsurface water. Therefore, many uncertainties persist in the model simulation (Lian et al., 2020; Liu et al., 2021a). The InVEST model requires two main input datasets, namely climate and LULC. These datasets may be prone to errors due to various uncertainties (Hoyer & Chang, 2014). Classification of landscapes contains subjective elements such as a lack of prior knowledge which may affect classification accuracy. Poor remote sensing image quality may also contribute to poor classification accuracy. These sources of uncertainty affect any ecological assessment (Hou, Burkhard & Müller, 2013). Precipitation data and evapotranspiration data were obtained through interpolation. In contrast, climate change is complex and difficult to simulate. Therefore, it difficult to obtain accurate meteorological data with high temporal and spatial resolutions. Model parameters, such as the evapotranspiration coefficient and maximum root depth, are derived from empirical data. Therefore, the accuracy of these data will affect the accuracy of model simulation to a certain extent, although the basic pattern of water yield should remain unchanged. Furthermore, although water yield is closely related to socioeconomic development and human activities, socioeconomic data and survey data are rarely used as model input data, thereby contributing to a certain degree of uncertainty (Lang, Song & Zhang, 2017). Despite some limitations, the InVEST model is widely regarded as a suitable tool for the evaluation of a variety of ecosystem services within catchments characterized by low data availability (Vigerstol & Aukema, 2011). Future research should conduct further field sampling in Shangri-La City to verify the water yield simulated by the InVEST model in the current study. On the other hand, previous studies have shown that the relatively simple InVEST water yield model can provide an accurate estimate of water yield provided that model input data and related parameters can represent the relevant spatiotemporal scales (Redhead et al., 2016).

Shangri-La City falls within a region containing a fragile and sensitive ecosystem. The region is also a valuable tourist destination in China. However, the development of tourism may result in various negative effects. For example, Shangri-La City is known for its beautiful scenery and attracts a large number of tourists to walk, rest, camp and barbecue on the grassland, whereas trampling on the grassland, throwing away garbage and pouring soda at will, all of which cause different degrees of damage to the grassland (Yang et al., 2019b). In order to further attract tourists and improve the hospitality capacity of the scenic spot, Shangri-la City has increased the construction of roads in the scenic spot. Due to the complexity of the geographical location and terrain of the study area, the process of road construction may damage the ecological environment on both sides of the road, resulting in the loosening and instability of mountains, thereby exacerbating the occurrence of geological disasters such as collapse, landslide and debris flow. In addition, future tourism development will also result in changes to the spatial distribution of land use types. Therefore, assuring sustainable future development of tourism and the economy requires a focus on intensive use of land and reasonable planning. In addition, there should be a move to transformation of traditional tourism to ecotourism to minimize the impact of land development on mountain water yield services.

Conclusions

Water yield plays an extremely important role in ecosystem services and is of great significance to regional ecological security and the sustainable development of natural resources. This present study used various datasets as input data for the InVEST water yield model to quantitatively evaluate the water yield ecological service function of Shangri-La City from 1974 to 2015. In addition, the importance of water yield was determined and the responses of water yield to various factors were analyzed. The main conclusions of this study are outlined below: Annual water yield in Shangri-La City is high in the north and low in the central region. The average water yield of the study area exceeded 100 mm. High water yields were mainly identified in Hutiaoxia, Jinjiang, and Shangjiang. Over the study periods from 1974 to 2015, water yield mainly initially increased and then decreased, with the smallest water yield of 199.06 mm occurring in 2015. The spatial and temporal variability of water yield emphasized the need for additional measures to improve the quality of the ecological environment, thereby strengthening the protection of water resources.

The ranking of grades of water yield ecosystem services in Shangri-La City according to the proportion of the study area was Grade II (43.36%) > Grade III (20.90%) > Grade I (20.22%) > Grade IV (15.52%). Areas showing Grade IV water yield were identified to the west of Jiantang, central Xiaozhongdian, central Gezhan, northwestern Dongwang, and the Hutiaoxia. These areas should be given sufficient attention and priority protection.

The water yield ecosystem service functions are affected by precipitation, evapotranspiration, vegetation types, and topographic factors. There was a positive correlation between water yield and precipitation, with this relationship shown to be significant in most areas. In contrast, there was a negative correlation between water yield and potential evapotranspiration. The effect of shrubs on water yield exceeded that of meadows and forests under different vegetation types. The average water yields initially decreased and then increased with increasing elevation. Average water yields initially decreased and then gradually increased with increasing slope. In general, the model parameters used in the current study can provide a reference for related research in regions under similar climate conditions.

Supplemental Information

Supplemental Information 1 Soil data used to calculate PAWC.

Click here for additional data file.

Supplemental Information 2 Average water yield, precipitation, PET, soil depth, and PAWC in each region.

Click here for additional data file.

Additional Information and Declarations

Competing Interests

Author Contributions

Data Availability

The authors declare that they have no competing interests.

Yuanhe Yu conceived and designed the experiments, performed the experiments, prepared figures and/or tables, and approved the final draft.

Xingqi Sun conceived and designed the experiments, performed the experiments, prepared figures and/or tables, and approved the final draft.

Jinliang Wang conceived and designed the experiments, authored or reviewed drafts of the paper, and approved the final draft.

Jianpeng Zhang analyzed the data, authored or reviewed drafts of the paper, and approved the final draft.

The following information was supplied regarding data availability:

The raw data is available in the Supplemental Files.

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
