# Peer review of "Using InVEST to evaluate water yield services in Shangri-La, Northwestern Yunnan, China"

_PeerJ, doi:10.7717/peerj.12804_

## Round 0.1 · original submission · Major Revisions

I have now received back the comments of two reviewers on your manuscript. Both of them were very positive and only made minor comments on the pdf attached. I'd kindly ask you to refer to the files while preparing your revised version of the manuscript to resubmit.

The paper is indeed very well written and the question addressed along with the method are straightforward.

I'd add a few more comments:

1) try to add a few more 'canonical' citations about ecosystem services;

2) Figs 6, 7, and 8 can be grouped into a single plate in a figure.;

3) In Fig. 1 it'd be best to have an inset map of China showing the location of the study region, but please, remember to use internationally-recognized boundaries and territories of China.

4) the paragraph mentioning previous uses of the InVEST model L. 85-104 should be reduced, you don't need to mention and explain all those papers. Select a few and focus on those only.

5) consider depositing the data used in the paper in a public repository as other people could reproduce your study, as per PeerJ guidelines;

6) cite all software packages appropriately;

7) provide more details on how soil data were collected;

8) cite the webpage where the land use data is available from NatureConservancy;

9) please, prepare a plot showing the temporal trend in *all* the variables used in the model (e.g., land use classes, etc) and also water yield, it's hard to see those changes by only looking at the maps;

10) do not cite figures in the Discussion.

And finally, I think the study would benefit very much from a time series analysis to test the correlation between changes in variables through time.

I have made other comments in the pdf attached.

·

Basic reporting

English can be improved in some parts of the text by using synonyms instead of repeating them.

Authors could be more eclectic in their references.

Notes about the figures are in the PDF annotations.

Experimental design

Há detalhes deixados para trás nos materiais e métodos e estão nas anotações do PDF.

Validity of the findings

no comment

Additional comments

Understanding which factors affect water yield in a sensitive and ecologically important region is a very interesting and pertinent subject, especially in the current moment of convergence of the effects of climate change.

The study is simple and easily replicable to other locations, which is relevant to science. Furthermore, it shows that, although there are weaknesses and uncertainties in InVEST, it can indeed be used to quantify, temporally and spatially, the water yield in a given territory.

Additionally, it is also interesting to understand what to do with the information generated in terms of the management and conservation of the territory in the study area. The issue of tourism and the prioritization of protection in areas with high water production was briefly explored. In this sense, a window of opportunity opens to be explored within the context of the present work, praising not only the use of InVEST but also the use of its outputs.

Reviewer 2 ·

Basic reporting

Water yield is an important issue in ecosystem services. This paper uses several input data layers for the Integrated Valuation of Ecosystem Service Tradeoffs (InVEST) model to estimate the water yield of Shangri-La City in Southwestern China from 1974 to 2015, and obtained important results. The paper is well written in general, and I have added my comments/corrections in the PDF document. The paper has a clear structure, sufficient literature references, and relevant discussions. The figures are also of high quality. I suggest that the authors add a small map (location of the study area in China) to Figure 1 so that readers can easily understand the geographical location of the study.

Experimental design

The experimental design is appropriate.

Validity of the findings

The conclusions are well stated. The limitations are also discussed.

Annotated reviews are not available for download in order to protect the identity of reviewers who chose to remain anonymous.

---

## Round 0.2 · Minor Revisions

I have received two reviews back. One of them pointed out that one aspect of the discussion needs to be reformulated.

From my own reading, I think the paper improved considerably and I thank the authors for preparing such a quick and consistent revision

·

Basic reporting

No comment

Experimental design

No comment

Validity of the findings

Just one point regarding the impacts of tourism on the topic of discussions does not seem consistent to me. More details in the attached .pdf.

Reviewer 2 ·

Basic reporting

The authors have addressed my comments and the other reviewer's comments. I think the paper can be accepted for publication.

Experimental design

The experimental design is appropriate.

Validity of the findings

The findings are valid.

Additional comments

None.

---

## Round 0.3 · accepted · Accept

All the required changes were made and I’m glad to recommend your manuscript for publication as is.